# Transfer Bandwidth Optimization for Multichannel Time-Correlated Single-Photon-Counting Systems Using a Router-Based Architecture: New Advancements and Results

**Andrea Giudici \***, **Giulia Acconcia**, **Francesco Malanga** and **Ivan Rech**

Dipartimento di Elettronica, Informazione e Bioingegneria—Politecnico di Milano, Piazza Leonardo da Vinci 32, 20133 Milano, Italy; giulia.acconcia@polimi.it (G.A.); francesco.malanga@polimi.it (F.M.); ivan.rech@polimi.it (I.R.)
**\*** Correspondence: andrea.giudici@polimi.it

**Abstract:** Time-correlated single-photon counting (TCSPC) is a powerful technique for time-resolved measurement of fast and weak light signals used in a variety of scientific fields, including biology, medicine, and quantum cryptography. Unfortunately, given its repetitive nature, TCSPC is recognized as a relatively slow technique. In the last ten years, attempts have been made to speed it up by developing multichannel integrated architectures. Yet, for the solutions proposed thus far, the measurement speed has not increased proportionally to the number of channels, reducing the benefits of a multichannel approach. Recent theoretical studies and prototypes have shown that it is possible to implement a new multichannel architecture, so-called router-based architecture, capable of optimizing the efficiency of data transfer from the integrated chip to the data processor, increasing the overall measurement speed. However, the first implementations failed to achieve the theoretical results due to implementation flaws. In this paper, we present a new logic for the router-based architecture that can operate at the same laser frequency and solve the issues of the previous implementation. Alongside the new logic, we present a new integrated low-jitter delay line combined with a new method for timing-signal distribution that allows the proper management of the pixel timing information. The new implementation is a step closer to realizing a router-based architecture that achieves the expected theoretical results. Simulations and bench tests support the results here reported.

**Keywords:** in vivo; imaging; multichannel; delay line; photon; router; SPAD; TCSPC

## 1. Introduction

In recent years, time-correlated single-photon counting (TCSPC) has become a fundamental technique for many scientific, medical, and industrial applications where it is necessary to measure fast and faint luminous signals with picosecond resolution [1]. In particular, TCSPC is applied to remote sensing, like light detection and ranging (LIDAR) [2,3], while in life science it enables high-precision analysis such as fluorescence lifetime imaging microscopy (FLIM) [4–6] and Förster resonance energy transfer (FRET) [7–10]. In a typical TCSPC measurement, a sample is excited by a pulsed laser source, and a histogram with the same shape as the optical curve is constructed by measuring the arrival time of each re-emitted photon [11]. To avoid the distortion of the recorded curve caused by the so-called pile-up effect, the average number of detected photons per period is typically limited between 1% and 5% [11]. Furthermore, many acquisitions are required to correctly build the histogram: consequently, TCSPC is acknowledged as a relatively slow technique compared to other imaging techniques, such as the time-gated [12].

One solution to speed up a TCSPC measurement consists in parallelizing the number of acquisition channels, moving from a single channel to a multichannel architecture. Numerous multichannel systems have been proposed in the literature to pave the way for

consistent measurement speedup [13–15]. The advancements in CMOS technologies have enabled the development of multichannel systems comprised of linear or two-dimensional arrays of single-photon avalanche diode (SPADs) photodetectors and the associated electronics. SPAD-based systems have gained prominence due to the detector's excellent performance [16] and ease of integration, which has enabled the development of systems with thousands of channels [13,17,18]. Given the high number of channels of these systems, crosstalk problems can arise, such as optical and electrical crosstalk between adjacent channels [19], as well as problems of saturation of the transfer speed towards the processing unit [20]. While the first category of problems limits parameters such as fill-factor and the physical dimensions of the system, the second one limits the amount of information transferred out of the chip, limiting the system's speed. Multichannel systems typically use "clock-driven" or "event-driven" architectures to extract information from the integrated chip [13,21–26]. Clock-driven architectures utilize a reference clock that sequentially scans and reads each pixel. An integrated conversion circuit, typically a time-to-digital converter (TDC), generates the timing information in the pixel and stores it in a local register. The TDC in a TCSPC system is a critical component of the entire system, as the system's resolution and precision are heavily reliant on the conversion circuit. The presence of a TDC per pixel imposes significant space and power constraints, establishing a trade-off with resolution and precision. The local register, on the other hand, is read by the system only after the entire system has been scanned, therefore keeping the pixel busy until the scan ends, resulting in inefficient resource use. Indeed, in large arrays with thousands of pixels, the sequential readout can take several microseconds, restricting the maximum working rate of each pixel in the Mcps range [22,24,25]. On the other hand, event-driven architectures subdivide the system into clusters of pixels, in which the timing circuit is shared among the pixels of the cluster, relaxing its power and area constraints. The resource-sharing technique is a critical parameter in this architecture because it impacts the overall efficiency of the system. For example, each time a photon is detected by a pixel within the cluster the shared timing circuit performs the conversion and the whole cluster enters a dead time set by the timing circuit [21]. Large dense SPAD arrays can theoretically produce large amounts of data, which can easily reach 100 Gbit/s. As an example, consider an 80 MHz sample excitation frequency, which is typical for mode-locked lasers used in medical and biological applications, and a moderate 5% photon detection rate to avoid pile-up distortion. If every pixel is equipped with a TDC and two bytes are used to encode the timing information, it results in a 64 Mbps data rate per pixel. If we consider a $32 \times 32$ pixel system, the data rate could theoretically be as high as 65.5 Gbps. Furthermore, if the application requires pixel-location capabilities (10-bit address for a $32 \times 32$ array), the previous value rises to 106.4 Gbps. Unfortunately, such high-speed real-time handling necessitates a huge bus bandwidth to the external processor, a high number of I/O pads, as well as significant system complexity. As a result, bandwidth saturation is currently one of the primary constraints on the speed of TCSPC measurements.

Different solutions have recently been proposed in the literature to deal with the bus's limited bandwidth to the external processor, attempting to maximize its utilization under typical TCSPC operating conditions. On one hand, the transfer rate can be increased at the chip level by leveraging data compression, such as histogram generation directly on chip [27,28], or by lowering the resolution [29]. As a result, a larger amount of data is extracted, but at the expense of performance (e.g., timing precision). To optimize the use of the transfer bandwidth without compromising other performance, Cominelli et al. in [20] proposed a new type of architecture, called "router-based". The idea driving this new architecture is starting from the bottleneck of the system, i.e., the limited transfer bandwidth, and determines the number of time converters required to achieve a bandwidth sustainable by the system in terms of I/O connections and power request. Assuming a time converter that can operate at the same excitation frequency, i.e., without dead time, two bytes to encode the timing information, and 10 address bits to locate the pixel within the array, four time-converters and 1024 pixels are enough to reach a transfer bandwidth of

~10 Gb/s, sustainable by a compact system characterized by low power operation and a limited set of I/O resources. In this scenario, the role of the router-based architecture is to act as an intermediary between a large set of pixels and the limited set of time converters, guaranteeing a dynamic and fair association between the two sets. The main features of the router-based architecture can be summarized as follows: (i) the data transfer from the chip is synchronized to an external reference signal that is the laser clock, regardless of the number of pixels to be read out; (ii) the external timing circuits are dynamically and fairly shared among all pixels, thus providing efficient use of resources and no bias in the readout; (iii) the exploitation of a limited number of external timing circuits paves the way to their timing performance optimization. For example, it is possible to use a time-to-amplitude converter (TAC) that, at the expense of area occupation and power dissipation, currently provides state-of-art picosecond precision, sub-picosecond resolution, and low differential non-linearity (DNL) [30]. Given N pixels and M timing resources, the architecture can randomly choose which pixels to associate with the timing resources, for each excitation period. The architecture must guarantee the absence of bias in the choice of the pixel: for instance, it must allow a random selection rather than always selecting the pixels that are triggered first throughout the excitation phase to prevent distortions. The maximum transmission bandwidth between the chip and the main processing unit (FPGA) determines the number of pixels N and the number of timing resources M.

Cominelli et al. in [31] proposed the first digital implementation of the router-based approach, consisting of a smart routing logic able to effectively connect a large detector array to a limited set of external time measurement circuits. Although that solution proved the feasibility of the architecture, it coule still be optimized to extract a large amount of data while ensuring improved performance. Indeed, from an implementation point of view, that structure requires a 400 MHz clock to synchronize its different stages, which poses a challenge in the design of the electronics in terms of disturbances and power dissipation. Moreover, from an application point of view, that structure suffered from a counting-loss effect at low rates of impinging photons, which limits the effective measurement speed below the one calculated for a theoretical router-based approach.

In this paper, the technical improvements we have made to the router-based approach are shown. In particular, the new architecture works in sync with the excitation rate. Another advantage of the proposed architecture is a higher measurement speed compared to what was obtained with the previous algorithm. In particular, the efficiency of the implementation described in this paper coincides with the one calculated in [20] for a theoretical router, which is the maximum achievable efficiency for a router-based approach. A 150 nm technology node was used to design, simulate, and implement the improved architecture. The architecture has been separated into multiple blocks based on their specific function, making it easier to evaluate the performance of each individual block.

This paper is organized as follows: in Section 2, the implementation of the system is described; in Section 3, the experimental setup and the obtained results for each block are described. Conclusions are drawn in Section 4.

## 2. Router Structure and Working Principle

For each laser excitation period, the router-based architecture can dynamically associate N pixels with M time converters. The architecture consists of a logic circuit replicated in each pixel, a selection logic, and a logic dedicated to the extrapolation of the timing signals from the chip. The latter two are shared by all pixels and distributed within the structure, while the pixel's logic regulates the temporal operation. The router's temporal flow is depicted in Figure 1, with the operating phases of the router associated with each excitation period of the laser. The first excitation cycle is used as a dwell phase, during which the pixels are active and ready to receive photons. When a pixel is triggered, it joins all the pixels that have been triggered during the same dwell phase in the selection process. The router decides which pixels to route to the time converters during the selection phase. To avoid measurement distortions, the selection process is unbiased. Since the selection

requires some time, the pixel's timing information is temporarily stored in its delay line. Once the pixel is selected, the router dynamically generates the path that connects the pixel's delay line to the selected converter during the Tim_path phase and generates the pixel's address. In the end, during the Conversion phase, the timing signal propagates towards the converter. The four phases just described occur in synchrony with the laser's excitation frequency, 80 MHz, which translates into a fixed duration of each phase equal to 12.5 ns. To maximize the system throughput, a pipelined structure has been implemented: in this way, when some pixels are using the external converters (Conversion phase), the system can already choose the pixels that will gain access to the external resources at the next round, thus producing output data at each laser period.

**Figure 1.** Timing diagram of the routing algorithm. Each time a photon is detected during a dwell time a selection process starts, lasting for one period. At the end of this phase, the router dynamically generates the timing path, and the delayed signals are measured by means of the external converters during the Conversion phase. Each excitation period can be exploited as dwell time, thanks to a pipelined mechanism.

It is worth noting that the router-based architecture is not dependent on the timing information coding methodology: it can be used with both TDC-based and TAC-based architectures. The analog or digital coding has no effect on the selection algorithm. The performance of the two converters, TDC or TAC, drives the choice between the two approaches. When comparing parameters such as precision, DNL, and FSR, TACs currently offer state-of-the-art performance [30] at the expense of high area and dissipated power. Our decision to use an analog approach with external TACs manufactured in a mature 350 nm technology node allows us to benefit from its state-of-the-art performance, and moves a critical element such as the time converter outside the chip containing the pixels, easing area, and power constraints.

The overall architecture is shown in Figure 2. The individual logic blocks are discussed in the following subsections, both operationally and in terms of implementation.

### 2.1. Pixel Logic

The logic of the pixel, schematically represented in the top part of Figure 2, can be divided into five blocks based on the function performed by each one of them. The finite state machine (FSM) is the first examined block, followed by the delay line and its demux, and finally the calibration logic. The quenching circuit block will not be examined as it is based on the structure described in [32].

When a photon triggers an avalanche in the SPAD, the time-resolving quenching circuit intervenes and produces two signals at its output: a digital pulse and a timing signal. While the first one simply points out a photon detection event, the second one allows the extraction in the photon arrival time with picosecond precision. These two signals are routed through calibration mux, the timing signal to the delay line and the digital pulse to the pixel state machine. The calibration mux is only used during the calibration phase: it allows direct access to the delay line and state machine inputs with deterministic signals to characterize the entire structure. Finally, the state machine communicates its status with the selection logic and receives from it the information about which converter and routing

path to use. After obtaining this information, the state machine applies it to the demux via the "sel" command.

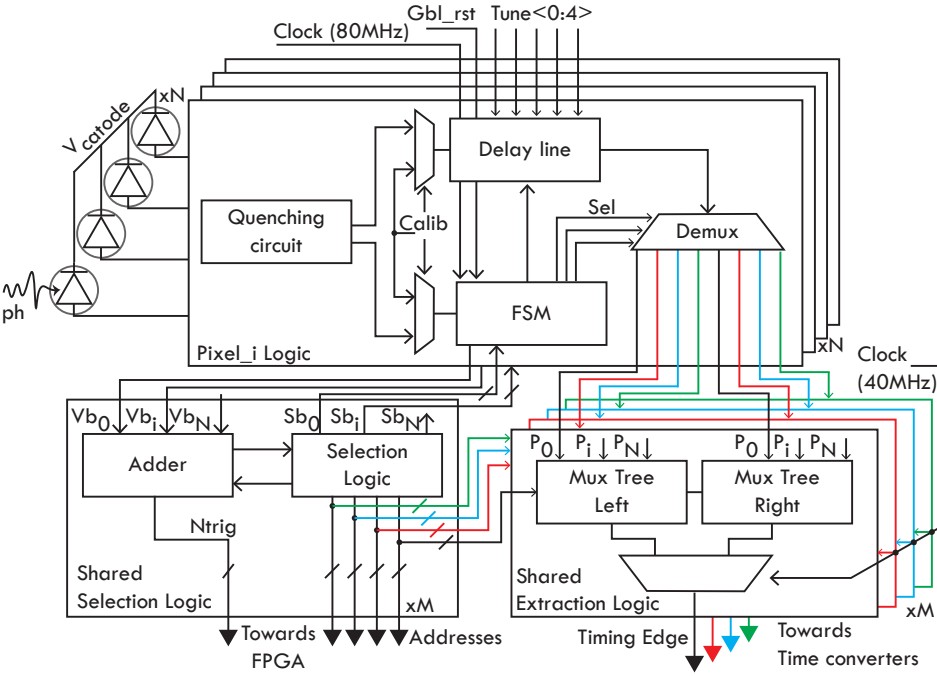

**Figure 2.** Schematic representation of the architecture. Each pixel communicates with the shared section logic, which, based on its choice, communicate to the shared extraction logic which pixel route to the external time converter.

The individual blocks will be examined in detail in the following subsections.

### 2.1.1. Finite State Machine

Each pixel has a finite state machine that regulates the pixel's operation and interaction with the selection logic and the delay line. The FSM schematic and state diagram are shown in Figure 3. The state machine starts in a dwell phase in which the pixel is active and ready to receive a photon. Once the pixel receives a photon ($Ph = 1$), the state machine registers the event, and at the beginning of the selection phase triggers its output validity bit ($Vb$) to participate in the selection mechanism. If during the selection phase, one of the four selection bits ($Sb_x$) coming from the selection logic goes high, then the FSM enters the Tim_path phase; otherwise, the FSM goes back into the dwell phase and it resets the delay line via $Rst\_Dly$. Furthermore, if the pixel has been selected ($SB_x = 1$), the FSM generates accordingly the $SEL\_M_x\_L/R$ bit for the pixel demux. The $SEL\_M_x\_L/R$ bit will be used during the Tim_path phase to route the delay line's output to the correct converter ($M_x$) via the correct path ($L$ or $R$). During the Tim_path phase, the router generates the timing paths, while the FSM takes no action. In the end, during the conversion phase, the timing signal from the delay line propagates towards the external converter, and the FSM returns to the dwell phase, ready to receive the next digitized photon.

### 2.1.2. Delay Line

The selection phase and the Tim_path phase have a finite duration equal in total to two clock periods, and it is necessary to preserve the pixel's temporal information for their whole duration. If this were not the case, the time information encoded in the rising edge of the timing signal would be lost. As a result, each pixel has a delay line that serves as an analog memory, preserving the timing signal's edge. If the pixel is selected, the edge propagates in the signal extraction logic toward the time converter during the Conversion phase; otherwise, it is discarded.

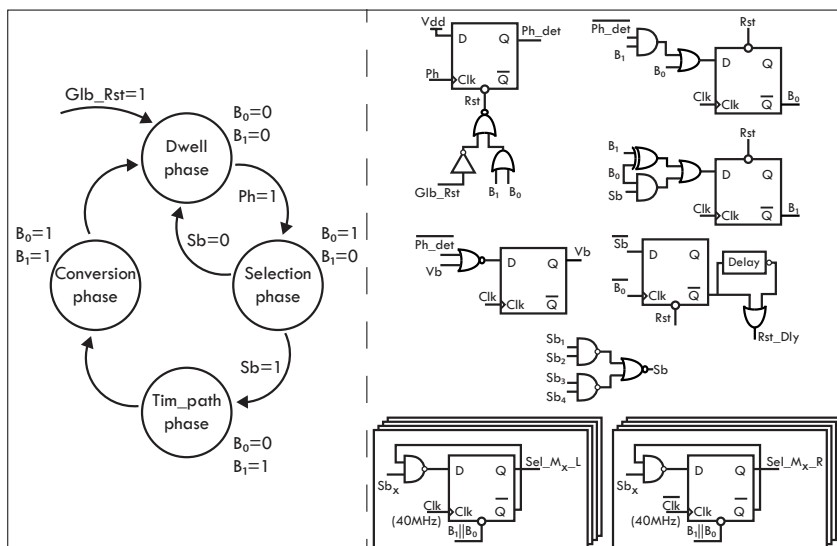

**Figure 3.** Finite state machine schematic: a two-bit FSM is implemented in each pixel.

The delay line is a critical component of the architecture because it is directly in the path of the timing signal. Typical delay line specifications include introduced delay and jitter, power consumption, and occupied area. An ideal delay line, in particular, must have a stable delay, combined with low power and low area consumption. To maintain a low timing precision of the entire system, we aim to minimize the timing jitter introduced by the delay line. However, a delay line must be included in each pixel of the array, limiting the maximum area and power dissipation, requirements that are typically in trade-off with jitter minimization. Thin custom-technology SPAD photodetectors typically feature a jitter of ∼30–35 ps full width at half maximum (FWHM) at room temperature [16], whereas TACs can achieve precisions down to few ps [30]. As a result, the delay line must have a low jitter, so that the dominant contribution of the system is approximately equal to that of the photodetector. If the application requires a high photon-detection efficiency (PDE∼40%) at 800nm, the RE-SPAD [33] can be used, which relaxes the jitter specifications by introducing a jitter of ∼100 ps FWHM.

Concerning the delay to be introduced, as shown in Figure 4, this must be equal to 37.5 ns; i.e., it must last for the entire dwell phase as well as the entire selection and Tim_path phase.

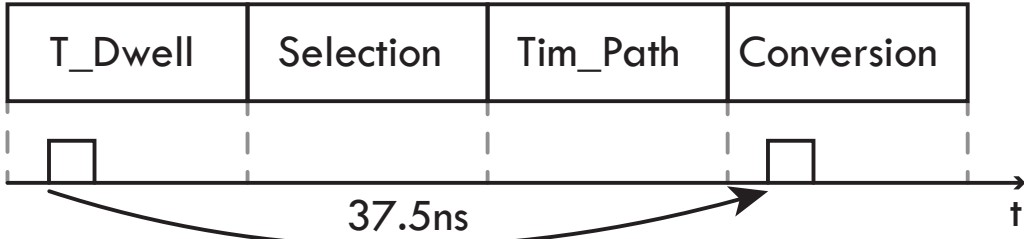

**Figure 4.** The delay introduced by the delay line must cover three phases, that is a delay of 37.5 ns.

It should be noted that the maximum delay attainable is in tradeoff with the jitter minimization [34]. In order to minimize the required delay, and thus the jitter, a compact design of the selection logic has been devised. This block is able to perform its task in just one clock period.

The presence of process variability and mismatches in the fabrication of the circuits contributes to the criticality of repetitive structures such as arrays. For this reason, it is possible to act externally on the delay introduced by means of a tuning bit to mitigate the effect of process variability. In the opposite way, in terms of mismatch variability, there is a risk

that in an array all the delay lines introduce different delays, necessitating the calibration of each delay line. The structure's design allowed us to limit the mismatch variability of the delay below the TAC's precision, greatly simplifying the calibration operation.

A variety of circuit types can be employed to implement a delay line [34,35]. To achieve the demanding combination of long delay, low jitter, and low power and area that a SPAD-based system for single-photon timing requires, we designed a mixed analog/digital structure. As shown in Figure 5, the analog block consists of a ring oscillator and an output stage, while the digital block consists of a counter and comparison logic.

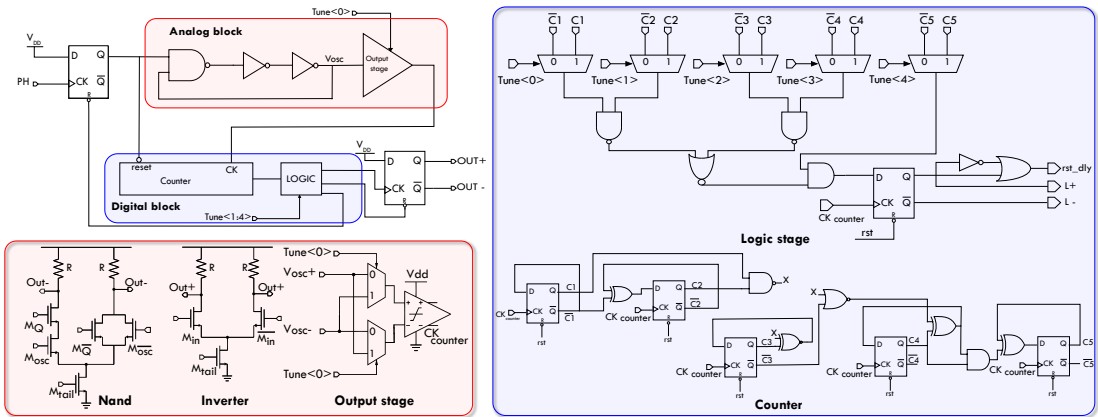

**Figure 5.** Delay line schematic. In red is highlighted the analog block, which is composed of a ring oscillator and an output stage. In blue is highlighted the digital block, which consists of a counter and a logic stage that checks when the counter reaches the desired value.

When a photon is detected, the flip-flop closes the ring oscillator's loop, causing it to oscillate. The oscillation feeds the digital part's counter, which counts the number of oscillations. When the counter reaches the number of oscillations specified by the tuning bit, the logic stage outputs the timing signal and resets the counter. An insight into the analog part is also visible in Figure 5. It is implemented with a differential architecture based on current mode logic (CML) ports. The use of differential signals makes the architecture more robust against disturbances, whereas the use of a constant current architecture reduces noise on the analog power supply. The oscillator has an 840 ps cycle-by-cycle period. Although pMOS transistors have less flicker noise, nMOSs were chosen because they have higher conductivity, allowing the input transistors to be smaller. This results in a twofold advantage: a lower overall area occupation, which favors integration density, and lower parasitic values, which translate into a faster and steeper transition, with beneficial effects in terms of jitter and noise immunity. The transistors' sizing is visible in Table 1: the tail nMOS has been sized to have a current value reasonable for an array structure (about 150 μA) while also ensuring low jitter. The whole circuit is powered at 1.8 V.

**Table 1.** Sizing of the inverter and the AND gate of the proposed delay line.

| INV | Value | AND | Value |
|:---:|:---:|:---:|:---:|
| $\frac{W}{L}$ $M_{input}$ | $\frac{8.5\ \mu m}{0.51\ \mu m}$ | $\frac{W}{L}$ $M_{Q,\overline{Q}}$ | $\frac{4.1\ \mu m}{0.45\ \mu m}$ |
| $\frac{W}{L}$ $M_{tail}$ | $\frac{20\ \mu m}{2\ \mu m}$ | $\frac{W}{L}$ $M_{Osc,\overline{Osc}}$ | $\frac{8.2\ \mu m}{0.45\ \mu m}$ |
| R | 4.5 kΩ | R | 5.4 kΩ |

The ring oscillator output is routed through an output stage, which converts the differential signal to a single-ended signal that can be used by the digital part. By means of the output stage, it is possible to choose whether to work on the rising or falling edge of the oscillating signal using the tune(0) bit. This allows for fine-tuning (420 ps) of the delay. To

reduce the jitter introduced by this stage, the input transistors have been sized to minimize their capacitive contribution to the speed of the internal nodes of the ring oscillator.

The digital block is made up of a synchronous digital counter clocked by the ring oscillator's signal. The counter result is compared to the logic block's tuning word and the timing output is propagated out of the delay line once the two values match. Figure 6 shows the post-layout simulated results: the jitter is around 8 ps FWHM at 37.5 ns delay. In terms of power consumption, the analog part is responsible for the majority of the static power contribution of 0.7 mW, with the ring oscillator contributing the most at 0.6 mW. The digital part, on the other hand, is responsible for the dynamic contribution up to 0.4 mW. The process variability of the delay is about 2 ns FWHM. This is within the delay line's tunability range and thus easily tunable. The mismatch variability, on the other hand, is contained within the ps, allowing all of the pixels' delay lines to be tuned in parallel, greatly simplifying the calibration process.

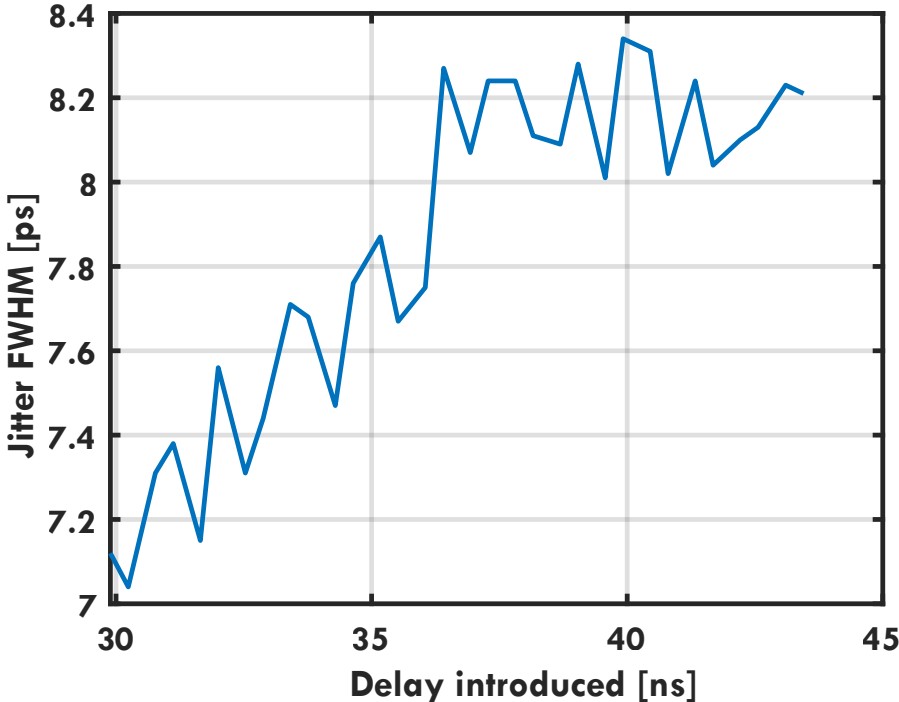

**Figure 6.** Post-layout simulated delay variability as a function in the delay introduced. On the vertical axis is shown the jitter expected.

### 2.1.3. Calibration Logic

The purpose of the calibration logic is to check the delay introduced by the delay line and the extrapolation logic and tune it using the tune <0:4> bit. Since the delay lines' mismatch is confined below the TAC precision, it is possible to use the same tuning word for all delay lines. In this case, the calibration logic consists of a simple demultiplexer that allows external access to the pixel, greatly simplifying the calibration process and the testing procedure.

### 2.2. Selection Logic

During the selection phase, all pixels that detected a photon (PH = 1) during the previous dwell period participate in the selection process (VB = 1), during which the selection logic randomly chooses a subset of pixels that gain access to the external converters. If the number of selected pixels is lower than the number of converters M, the number of selected pixels will be equal to the number of triggered pixels. Instead, only M pixels among those that have been triggered are randomly chosen to be routed towards the converters. If a triggered pixel is not selected, the selection logic discards it by turning its SB bit low

(SB = 0). Whenever a pixel is discarded, its finite state machine and delay line are both immediately reset, thus allowing the pixel to enter the dwell phase in the following cycle.

Let us consider Figure 7, which shows a high-level description of the structure of the selection logic. As can be inferred from the Figure the selection logic is divided into several blocks and distributed with a tree structure. The selection principle is divided into three steps, each of which is implemented by a separate circuit block. The first step is to figure out how many and which pixels are triggered during the dwell period. At the start of the selection phase, each pixel communicates its status, whether it is triggered or not, to a distributed adder (in red), whose job is to collect and add up the information on how many pixels in the array are triggered. This communication takes place using the validity bit (VB) generated by the pixels' FSMs. Since the number of time converters is limited, the sum produced by the adder is saturated and encoded in thermometric coding. This choice allows for a simpler adder structure, while thermometric coding allows for the assignment of a converter to each bit position in the sum word N_trig. So, if a structure is made up of four time-converters, the thermometric sum will be four bits long, with the LSB representing converter number one and the MSB representing converter number four. The second step begins after the sum word is generated, which is processed by the selection block (in blue) with the aid of the priority generator (in green). By knowing which and how many pixels are triggered, these two blocks communicate to the pixels via the selection bit (SB) whether they have been selected or not. The priority generator provides the selection logic with the ability to perform different selections at each excitation cycle. It is essentially a distributed counter, consisting of a flip-flop integrated into each node of the tree, which updates its output in sync with the laser frequency. Each selection block follows two fundamental rules: first, the overall number of high bits sent to the outputs is equal to the number of high bits contained in $SB_{in}$; second, if a bit in one output is high, the bit with the same position within the other output is '0'. This operation is performed on the basis of the priority bit, P, and the $\Sigma$ inputs: first of all, the priority bit is used to select a preferred direction, that is the left branch if P is equal to '0' or the right branch if P is equal to '1'. Then, once a direction has been selected, the logic exploits the $\Sigma$ input coming from that direction to send an equal number of high bit belonging to $SB_{in}$ to that part of the array. The remaining high bits are sent to the complementary output.

The address of a selected pixel contains a bit for each step of the selection process. In particular, given the binary structure of the tree, the number of steps is equal to the binary logarithm of the number of pixels, so the same holds for the number of bit composing an address. In general, the most significant bit (MSB) of an address is associated with the selection performed during the first selection step, while the least significant bit (LSB) is associated with the last selection step. In particular, the task of an address bit is to indicate if the associated selection step has sent a high selection bit to the left branch (address bit equal to '0') or to the right branch of the tree (address bit equal to '1'). Figure 8 shows the actual implementation of the single blocks composing the selection logic. It is worth mentioning that the complexity of the circuit architecture does not scale in relation to the number of pixels; rather, it scales with the number of time converters that are chosen after the available transfer bandwidth between the chip and the processing unit. The number of time converters dictates the length of the digital word N_trig, which represents the saturated thermometric sum of all pixels integrated over a specific time period. In the scenario detailed in the paper, the word N_trig will consist of four bits. The term "saturated" is used because there is no need to determine how many pixels exceeding the number of converters have been triggered.

With this approach, also the system's design complexity does not expand as the number of pixels increases. Once the shared selection logic node is reached at which the thermometric sum saturates (for instance, the second summing node in the case of four time-converters), all subsequent nodes are essentially duplicates of the same node. For a 32-pixel array, the number of shared selection logic nodes is equal to log_2(32), which is five nodes. Among these five nodes, when using four time-converters, the last three summing

nodes (the farthest ones from the pixels) will be identical to each other. In the context of a $32 \times 32$ architecture with four time-converters, there will be a total of 10 nodes, with the last 8 nodes having the exact same structure. As the number of pixels increases, there will naturally be a proportional increase in the occupied silicon area. However, the area actually taken up by interconnections and logic gates in the router-based architecture is smaller than that occupied by the pixel circuitry, like the quenching circuit and the delay line.

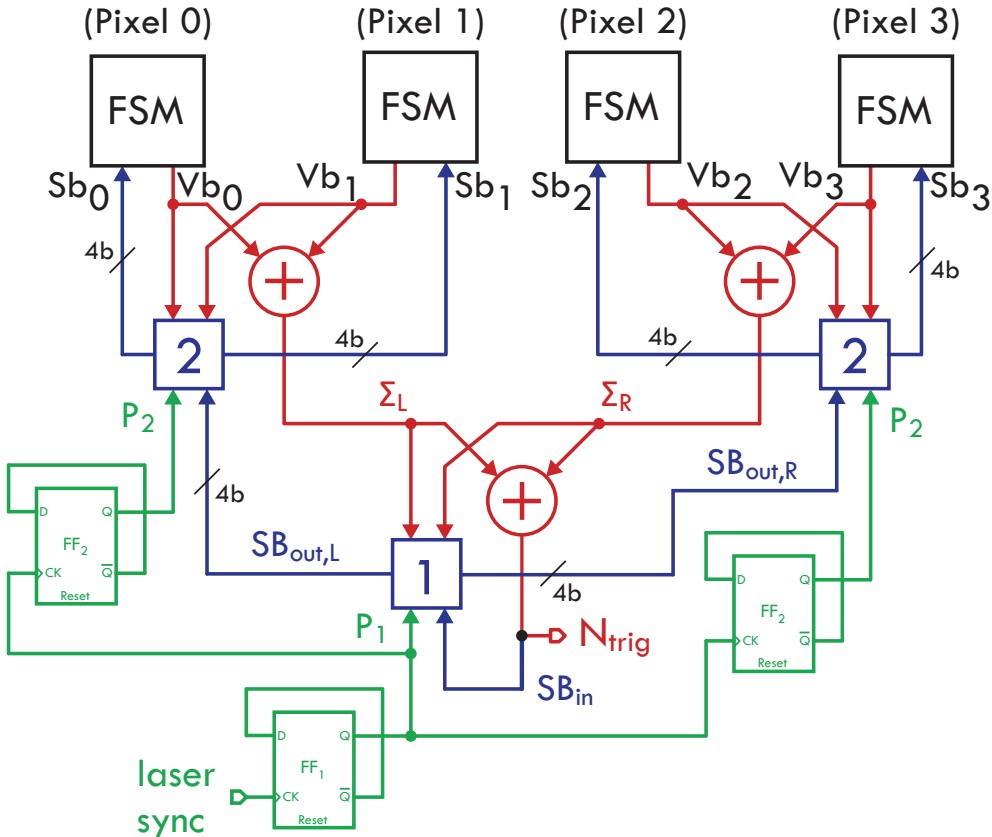

**Figure 7.** Architecture of the proposed routing logic for a $1 \times 4$ array. The in-pixel FSMs send a high-validity bit to a shared adder (in blue) every time a photon is detected. Then, after the addition phase has been completed, the result, $N_{trig}$, is sent to the output, and at the same time, it is fed to the selection logic (in blue). A flip-flop is integrated into each node to form a distributed priority generator, which is highlighted in green.

*2.3. Extraction Logic*

In order to extract the timing signal from the delay line and connect it to the various converters, it is possible, as a first approach, to connect the output of each delay line to each converter, via interconnections. This solution brings with it innumerable implementation criticalities. In fact, given an $N$ number of pixels and an $M$ number of time converters, the number of interconnections would be equal to $N \times M$. This large amount of interconnections (i) would be difficult to scale with the number of channels, (ii) would require a huge amount of I/O resources, and (iii) would require a converter capable of accepting all these connections. Such a static approach to the problem is therefore unfeasible in a large structure.

As a result, a dynamic solution has been proposed: the interconnections between the pixels and the converters are shared among the pixels, allowing each converter to service all the pixels. The word *tree* will be used from now on to refer to the group of interconnections that connect a time converter to all pixels. Given $M$ time converters, there will be $M$ trees, each uniquely associated with a converter. From a topological point of view, the dynamic solution does not present the same problem as the static approach, since it is true that the

number of converters may increase, but this increment will always be limited to a few units, while the number of pixels can vary by hundreds of units.

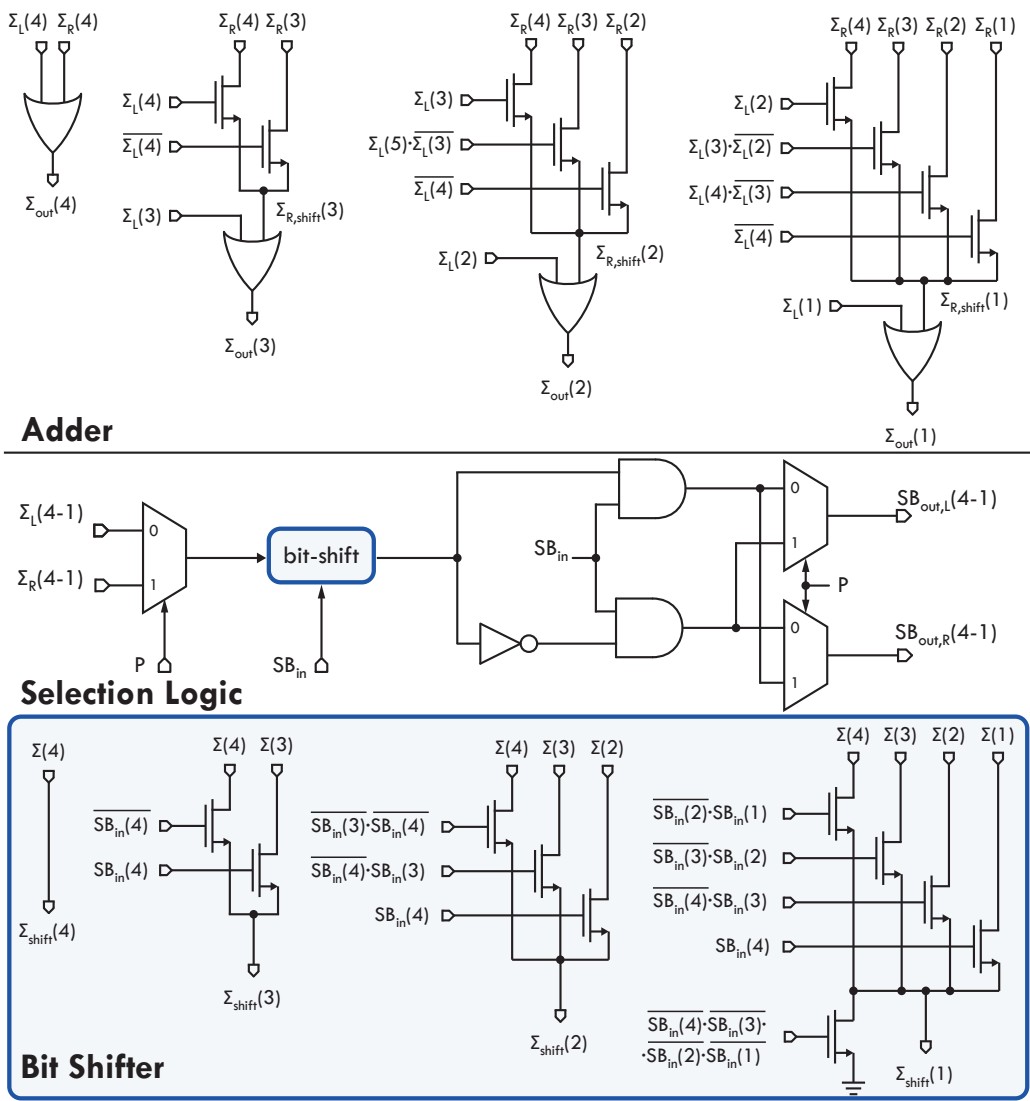

**Figure 8. Top**: Schematic of a generic adder integrated into a node of the tree. The circuit exploits four bits coming from the left branch, and four bits coming from the right branch, to generate a four-bit thermometric code. **Center**: Block diagram of the combinatorial electronics devoted to one step of the selection process. **Bottom**: Schematic of the bit-shifter block exploited in the selection logic. The logic shifts the input word, $\Sigma_{(4-1)}$, according to the position of the most significant high bit within the $SB_{in(4-1)}$ word. The result is $\Sigma_{shift(4-1)}$.

From a temporal point of view, i.e., the usage time of the interconnections in the static case, and of the tree in the dynamic case, present a criticality, regardless of any type of interconnection being characterized by a finite propagation time. In the case of a static connection, e.g., a metal track, the propagation time will be determined by the resistance and capacitance of the track, both also functions of the track length. In the case of a dynamic connection (tree), the propagation times of the logic gates that allow the sharing of the tracks will be added to the propagation time of the track itself. For multichannel structures, the propagation times can reach values of some *ns*. The propagation time is added to the delay introduced by the delay line and this causes all the pixels that detected a photon near the end of the integration phase to go beyond the conversion phase.

Figure 9 depicts a case in which the propagation delay is 2 ns and the delay line introduces an exact delay of 37.5 ns. Exceeding the conversion phase brings with it various

problems. First, the correlation between the number of triggered pixels (the word $N_{trigg}$) generated by the selection logic and the number of converters operating in the same phase is lost. This condition complicates data management on the processing unit side. In addition, a condition would be generated in which the pixels taken near the end of the integration phase would be treated as photons taken at the beginning of the next integration phase: a folding of the measurement would take place. This condition would further complicate the histogram reconstruction algorithm and would be increasingly aggravated as the propagation time of the extraction logic increases. For these reasons, the idea of compensating the propagation time can be taken into consideration as it is a deterministic quantity: a time offset. Ideally, this offset can be compensated for by reducing the delay introduced by the delay line by the same amount. However, this condition would collide with the selection logic timing: it is the selection logic that tells where the delay line must route the signal. The selection logic conditions the routing of the signal through the demux present in each pixel out of the delay line, as shown in Figure 2. The introduction in the demux prohibits the offset compensation of the propagation times, precisely because the demux is controlled by the selection logic, which enables the demux at the end of the selection phase.

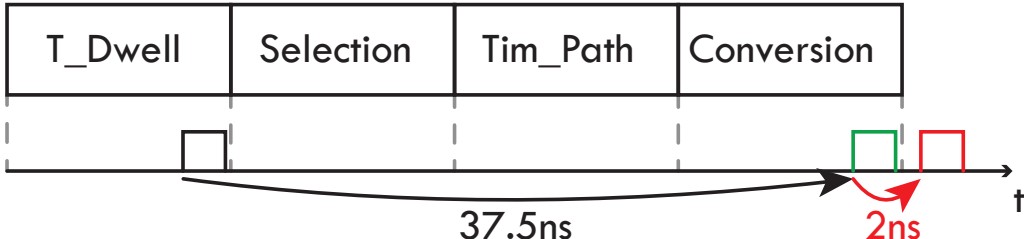

**Figure 9.** Example of conversion phase overrun. The delay line introduces exactly 37.5 ns, while the extraction logic processing time is equal to 2 ns, highlighted in red in the figure. This additional time results in the conversion phase being overrun, causing difficulties in the post-processing reconstruction algorithm.

Figure 10 provides a practical example of this concept. Assume that the propagation time of the tree is 2 ns and therefore is possible to compensate the delay introduced by the delay line by this value. It will therefore happen that a pixel triggered towards the end of the dwell period will actually reach the time converter after 37.5 ns. However, a pixel triggered in the first 2 ns of the dwell time will see the output of its delay line move after 35.5 ns, but here the selection logic has not yet made the decision on which pixel to select: the selection phase is still ongoing. The timing signal is therefore lost at the input of the pixel's demux because the pixel does not know where to route the signal yet. Therefore, propagation time compensation is not feasible with this implementation.

Compensating for propagation time becomes achievable through the utilization of the *Tim_path* phase. This phase enables the relaxation of time constraints attributed to propagation times within the extraction logic. Consequently, it permits treating the propagation delay as an offset, facilitating its compensation.

This solution brings with it a disadvantage, i.e., the reduction in the throughput of the architecture, as visible in Figure 11: during the *Tim_path* and the conversion phase, the extraction resources (the trees) are occupied by valid information and therefore cannot process the new information belonging to the following stage of the pipeline. In *Case1* the tree manages the signals coming from a triggered pixel and cannot handle the signal from another pixel triggered during the following dwell phase. To resolve this issue, it is possible to double the extraction resources for each converter.

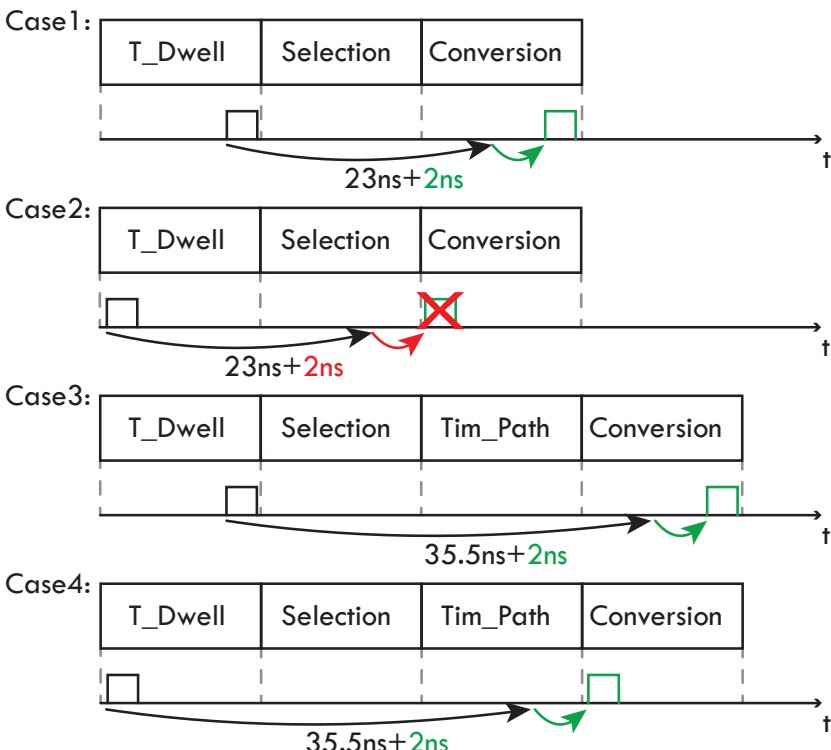

**Figure 10.** Representation of the problem that led to the introduction in the $Tim_{phase}$ phase in the state machine. In case 1, for a pixel triggered at the end of the dwell phase, it is possible to compensate for the propagation delay of the tree. In case 2, for a pixel triggered at the beginning of the dwell phase, compensation is not possible: the output of the delay line is activated during the selection phase and the demux does not know where to send the signal. The signal is lost. The introduction in a fourth phase, $Tim_{phase}$, dedicated exclusively to the management of propagation times, makes it possible to compensate for the offset in cases 3 and 4.

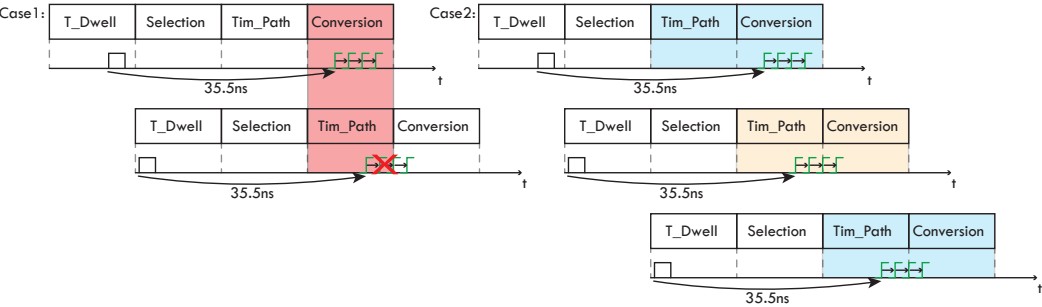

**Figure 11.** Example of how doubling the extraction resources keeps throughput constant. In case 1, two pixels triggered in two consecutive dwell phases of the pipeline require access to the extraction resources. These resources serve the pixel triggered in the first dwell phase and remain busy for a total of 25 ns, to ensure total signal transmission. This condition collides with the propagation of the timing signal of the pixel triggered during the second dwell phase. Yet, as shown in case 2, a doubling of the extraction resources, highlighted in light blue and light brown, allows both timing signals to be managed correctly: each resource processes its own information. To maintain constant throughput, the two extraction resources work in antiphase, at half the system frequency.

As shown in Figure 11, the two trees belonging to the same time converter manage the throughput by working in antiphase with each other and at half the frequency of the state machine (40 MHz).

Figure 12 shows the extraction logic structure for a system comprised of four external converters. The extraction logic dedicates two tree paths, left and right, for each external

converter. The timing signal coming from the pixel delay line reaches the demux integrated into the pixel logic. The demux, based on the results of the selection logic, enables one of its eight outputs, effectively routing the signal in one of the extraction logic trees. The selection bit of the mux implemented in each node is based on the address bit generated by the selection logic during the selection phase. It is worth noting that it is the selection logic that ensures that for each tree there is one and only one signal traveling that specific tree. If not, the two analog signals would interfere with each other and the information would become corrupted. The tree structure consists of log2(N) levels given N pixels, with the last stage directed towards the external converter.

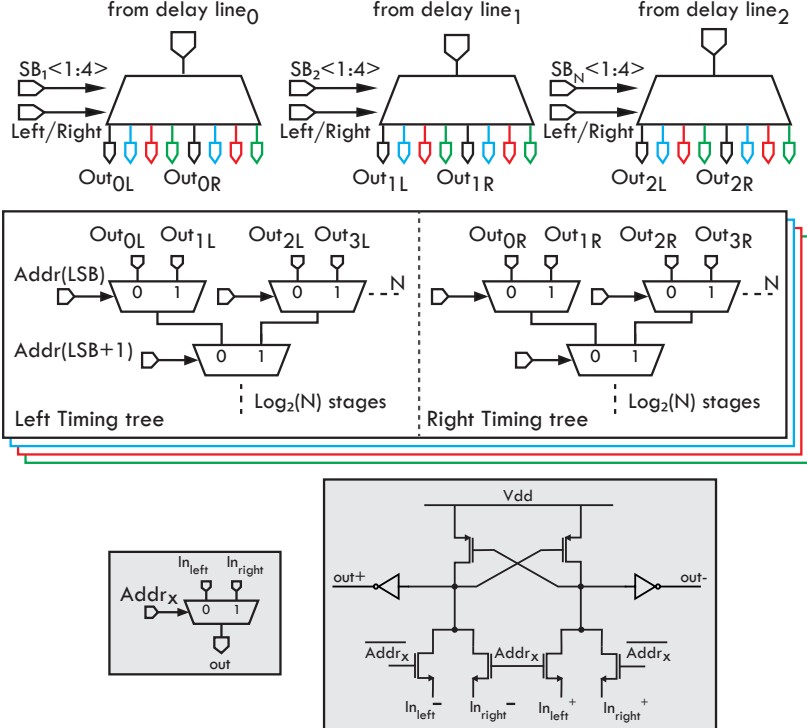

**Figure 12.** Schematic representation of the extraction logic. The output of every delay line reaches a demultiplexer that, based on the value of the select word and on the direction bit, links the delay line output to the timing tree. The two identical timing trees work in antiphase at 40 MHz.

## 3. Experimental Results

The new architecture has been fabricated using a 150 nm technology node. Figure 13 shows the individual circuit blocks. A portion of the overall architecture is highlighted in red, made up of 32 pixels and the selection logic; the extraction logic just described is currently being implemented. Two delay line prototypes are highlighted in blue: the prototypes share circuit similarities with the one described in the previous section. The goal of these prototypes is to validate the post-layout simulation results with experimental measurements. In this respect, the two delay lines have different static power consumption. The "low-power" delay line statically dissipates 0.36 mW at 1.8 V, while the "high-power" delay line statically dissipates 0.7 mW at 1.8 V.

### 3.1. Router Characterization

In terms of selection logic, the chip selects 4 triggered pixels out of 64 pixels. The 64 pixels were obtained by cascading two 32-pixel chips in a master–slave configuration. We started by testing a single chip, so just 32 pixels. We used a Kintex 7 FPGA mounted on a commercial demoboard (Genesys2 by Digilent) to conduct the tests. We designed an interface board that can handle the large number of interconnections that exist between chips and FPGAs. The test setup is visible in Figure 14. We developed three macro code

blocks in FPGA: (i) a logic function capable of replicating the selection logic function, (ii) a signal generator that will be used as input to the chip and block *i*, and (iii) a comparator capable of generating flags whenever the comparison between the outputs of block *i* and those of the chip is incorrect. The signal generator block allows to test the chip under various conditions. First, we generated a few signals and validated the chip's operation. Following that, we fed both the master and the slave chip all possible 32-input combinations and compared their outputs to those generated by the FPGA-implemented block. Given the high clock frequency (80 MHz), comparing all the combinations took only a few minutes. Finally, we provided input signals that were uncorrelated in time with respect to the logic operating clock in order to simulate random SPAD triggering. In all these situations, the selection logic met the needed requirements.

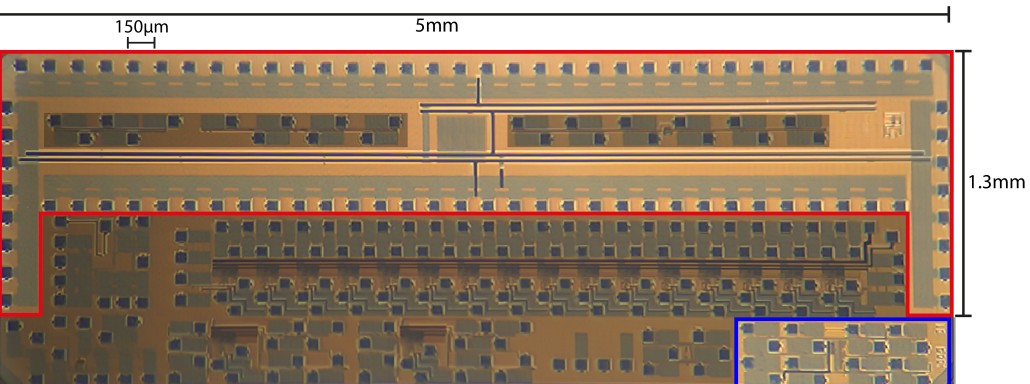

**Figure 13.** Chip image. In red is highlighted a portion of the router architecture with a pitch of 150 μm, while in blue are the first delay line prototypes.

### 3.2. Delay Line Characterization

We performed tests on the delay lines to validate the simulated results. To conduct the tests, we designed a board with particular attention paid to the component selection in order to reduce the jitter introduced on the signal path. At the board input, we generate pulses clocked at 4 MHz (5% of the laser frequency) using a pulser (Philips PM 5786B). The pulse fed to the board is split in two: one part goes to the delay line's input, while the other one is conditioned and transformed in NIM standard. The same network that conditioned the other signal also conditions the delay line output. The two output signals, separated in time only by the delay line's value, are fed into a commercial module (Becker&Hickl SPC130) [36] capable of measuring temporal distances with extreme precision. The prototypes showed the same jitter-delay trend simulated, that is, an increase in the jitter as the delay increases. Furthermore, the low-power delay shows a higher jitter than the other one, confirming the trade-off between power and jitter [37–39]. The measured average jitter is around 31 ps FWHM for the "high-power" delay line, while around 58 ps FWHM for the "low-power" delay line. The average jitter measured is within a 10% variability with respect to the simulated one. This information gives us more certainty on the simulated results of the optimized delay line. Then we changed the delay introduced by the circuit by changing its tuning bit and fully characterized the introduced delay as shown in Figure 15. As can be inferred from the graph, the "low-power" delay line jitter slope is steeper than the "high-power" one. This condition is due to the fact that the "low-power" delay line exhibits a reduced current consumption in comparison to the "high-power" delay line. This situation results in an extended oscillation period, consequently leading to a higher delay for the same tuning bit. Consequently, the high-power delay line not only provides a lower jitter but also a better resolution with respect to the low-power one.

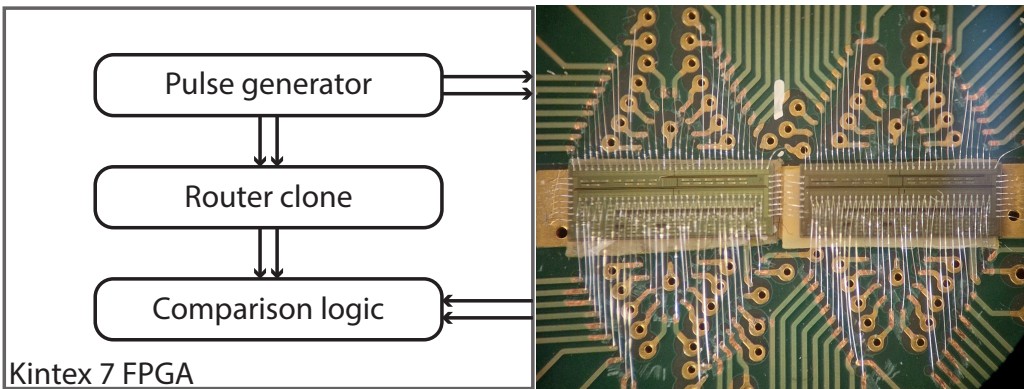

**Figure 14.** Test setup for the router characterization. The pulse generator implemented on FPGA feeds the input of the logic, emulating the presence of the SPAD + AQC. Then, the output of the selection logic is compared back into the FPGA with a clone block that mimics the selection logic behavior.

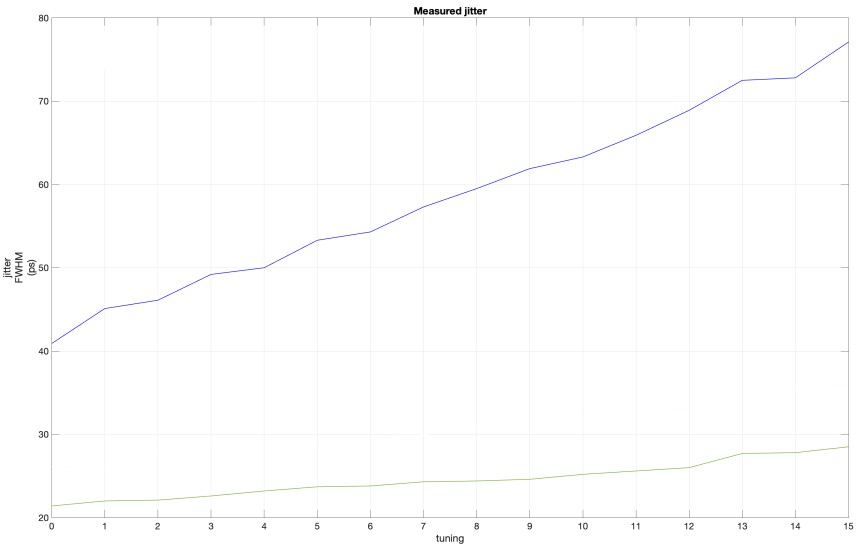

**Figure 15.** In blue is highlighted the jitter introduced by the "low-power" delay line, while in green the jitter introduced by the "high-power" delay line. Increasing the value of the tuning bit increases the delay introduced, and so the jitter is measured.

## 4. Conclusions

The new developments in the router-based architecture represent a step forward in the realization of a structure capable of ensuring efficient use of the transfer bandwidth between the chip and the processing unit without compromising key parameters of TCSPC acquisition systems, such as resolution and timing precision. If the router-based architecture ensures that the numerous pixels are correctly associated without bias with the limited high-performance time converters, it is up to the delay line and timing signal extraction logic to ensure that the time information reaches the converters, maintaining the highest level of fidelity. As a result, we designed a delay line and an extraction logic capable of influencing analog information as little as possible while also meeting the requirements of a multichannel structure, i.e., low power consumption and small dimensions. The positive test results of the individual circuit blocks lead us to the next step, which is combining all of the blocks into a 64-channel linear array. We believe that by combining this first-of-its-kind architecture with custom SPAD arrays and high-performance time converters, we will be able to accelerate TCSPC measurement while maintaining high timing performance.

**Author Contributions:** Writing—original draft preparation, A.G.; review, G.A., I.R. and F.M.; writing and editing, A.G. All authors have read and agreed to the published version of the manuscript.

**Funding:** This research received no external funding.

**Institutional Review Board Statement:** Not applicable.

**Informed Consent Statement:** Not applicable.

**Data Availability Statement:** Not applicable.

**Acknowledgments:** The authors would like to thank ex-students Gianluca di Carlo and Lorenzo Carbone (both currently at STMicroelectronics) for their contribution to the work.

**Conflicts of Interest:** The authors declare no conflicts of interest.

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
