# Peer review of "Transfer Bandwidth Optimization for Multichannel Time-Correlated Single-Photon-Counting Systems Using a Router-Based Architecture: New Advancements and Results"

_photonics, doi:10.3390/photonics10111227_

Round 1

Reviewer 1 Report

Comments and Suggestions for Authors

Item 1:

Talking about larger arrays of thousands of channels, like 32 x 32 arrays, the reader gets the impression that the proposed architecture would be of benefit to large arrays.  However, from the complexity of 32 pixels in the experimental results, it is difficult to gauge to which extend the proposed architecture could indeed serve in arrays of 32 x 32 and higher. 

Is the proposed routing logic becoming more and more complex in larger systems, hence the silicon area for that, becoming increasingly larger too, or not?

What is the expected maximum array size that this proposal could possibly serve, according to the authors?

Item 2:

The delay line that needs to delay the time carrying edge with good precision:  the proposed delay line is very circuit intense and power hungry, the question rises why hasnt been opted for charging a capacitor with a (settable) current source, followed by an Schmitt-trigger for delay generation ?  Circuit could be much smaller, maybe the variation on the delay is not good enough?

Item 3:

Figure 13:  can dimensions be indicated?  Like how wide is the cell, and how tall ?  In the caption it is mentioned that the pitch is 150nm, I assume this has to be 150 micron? 

Comments on the Quality of English Language

English is quite good, however, there are many places where the plural is not reflected in the noun: e.g. “four external converter”  instead of “four external converters” (line 388).  Line 391 “eight output”.   Etc…   it happens a lot, and normally, a spell-checker would show them.

Please correct.

Reviewer 2 Report

Comments and Suggestions for Authors

In the manuscript entitled ‘‘Transfer bandwidth optimization for multichannel TCSPC systems using a router-based architecture: new advancements and results”, the authors design a router-based architecture, including a delay line and an extraction logic, and the output and delay line tests of the TCSPC are performed. The work would be useful and related to multi-channel single-photon detection and laser source characterization [Optica 3, 1187 (2016); Optics Express 26, 5991 (2018); PNAS 112, 9258 (2015).] Therefore, I reckon it can be considered for a publication after the below mentioned concerns are clarified. (1) The experimental results should be presented in more detail, and what about the data transmission characterization? (2) The results of bandwidth optimization should be emphasized and clarified. (3) The authors show the resolution times of the "high-power" and "low-power" delay line, and what is the limit of the resolution time? Why is the "low-power" resolution time growing faster than that of "high-power"? (4) there are many typos that hinder reading. The following list is not an exhaustive list of language corrections. Line 17, typo in “allow”; Line 63, typo in “amount”; Line 70, typo in “rise”; Line 174, typo in “mux are”; Line 188, typo in “enter”; Line 195, typo in “propagate”; In Fig.12 caption, typos in “ouput”, “link”; Line 436, typo in “increase”; Line 443, typo in “characterize”.

Comments on the Quality of English Language

there are many typos that hinder reading. The following list is not an exhaustive list of language corrections. Line 17, typo in “allow”; Line 63, typo in “amount”; Line 70, typo in “rise”; Line 174, typo in “mux are”; Line 188, typo in “enter”; Line 195, typo in “propagate”; In Fig.12 caption, typos in “ouput”, “link”; Line 436, typo in “increase”; Line 443, typo in “characterize”.

Reviewer 3 Report

Comments and Suggestions for Authors

The paper raises the important topic of data transfer between the integrated circuit and its control system. The issue is valid as with the development of newer and faster integrated circuit the amount of data to be transfered increases drastically, yet, control systems do not keep up with this progress.

Line 27 - 29 - "To avoid the distortion of the recorded curve caused by the so-called pile-up effect, the average number of detected photons per period is typically limited between 1% and 5%" - what do you mean by "it is limited", who does limit it? Do you add some kind of attenuation?

Line 56 - I suspect the authors meant "pixels" instead of "detectors"? I find the term "detector" a little bit misleading, as previously you referred to pixels, so the reader can understand this phrase as "large arrays with thousands of detectors with thousands of pixels". I think more clarification would be useful here.

Line 278 - If the number of triggered pixels is greated than the number of converters, what happens to the rest, are they lost? I think the answer is yes, but if it is so, I suggest to state it more straight.

Also:

- What happens to the information from pixels that are waiting to be read out when new trigger comes before the architecture manages to read out the data? Is this scenario possible at all?

- Could you provide photon flux / bunch of photons interval when the system stops working as expected?

- How would the system behave in case of pile-up? 

I find the proposed method nice alternative to full-readout clk-based and event-based systems. 

Round 2

Reviewer 2 Report

Comments and Suggestions for Authors

The authors addressed most concerns and the paper is now more clear. Furthermore, for enhancing the significance of the work, the related optical and detection applications of the TCSPC should be complemented, and at least supplement some relevant and important references.

Comments on the Quality of English Language

The typos are corrected and the quality of english language is improved.

Author Response

Dear Reviewer,

Thank you for the revision.

Following your suggestion, I added the following citations with the aim to better clarify TCSPC applications.

Added citation N°3 line 23          

  • Lindner, S.; Zhang, C.; Antolovic, I.M.; Wolf, M.; Charbon, E. A 252 × 144 SPAD Pixel Flash Lidar with 1728 Dual-Clock 48.8 PS TDCs, Integrated Histogramming and 14.9-to-1 Compression in 180NM CMOS Technology. In Proceedings of the 2018 IEEESymposium on VLSI Circuits, 2018, pp. 69–70. https://doi.org/10.1109/VLSIC.2018.8502386.

Added citation N°5,6 line 24

  • Hirvonen, L.M.; Suhling, K. Fast timing techniques in FLIM applications. Frontiers in Physics 2020, 8, 161.
  • Datta, R.; Heaster, T.M.; Sharick, J.T.; Gillette, A.A.; Skala, M.C. Fluorescence lifetime imaging microscopy: fundamentals and advances in instrumentation, analysis, and applications. Journal of biomedical optics 2020, 25, 071203–071203.

Added citation N°8,9,10 line 25 :

  • Esposito, A. How many photons are needed for FRET imaging? Biomed. Opt. Express 2020, 11, 1186–1202..
  • Wang, S.; Shen, B.; Ren, S.; Zhao, Y.; Zhang, S.; Qu, J.; Liu, L. Implementation and application of FRET–FLIM technology. Journal of Innovative Optical Health Sciences 2019, 12, 1930010
  • Veerapathiran, S.; Wohland, T. Fluorescence techniques in developmental biology. Journal of biosciences 2018, 43, 541–553.

Added citation N°15 line 35

  • Gasparini, L.; Zarghami, M.; Xu, H.; Parmesan, L.; Garcia, M.M.; Unternährer, M.; Bessire, B.; Stefanov, A.; Stoppa, D.; Perenzoni M. A 32× 32-pixel time-resolved single-photon image sensor with 44.64 μm pitch and 19.48% fill-factor with on-chip row/frame skipping features reaching 800kHz observation rate for quantum physics applications. In Proceedings of the 2018 IEEE International Solid-State Circuits Conference-(ISSCC). IEEE, 2018, pp. 98–100

Reviewer 3 Report

Comments and Suggestions for Authors

I think that the paper can be published.

Author Response

Dear Reviewer,
thank you.